# Automated Identification of Overheated Belt Conveyor Idlers in Thermal Images with Complex Backgrounds Using Binary Classification with CNN

**DOI:** 10.3390/s222410004

**Published:** 2022-12-19

**Authors:** Mohammad Siami, Tomasz Barszcz, Jacek Wodecki, Radoslaw Zimroz

**Affiliations:** 1AMC Vibro Sp. z o.o., Pilotow 2e, 31-462 Kraków, Poland; 2Department of Robotics and Mechatronics, AGH University of Science and Technology, Al. Mickiewicza 30, 30-059 Kraków, Poland; 3Department of Mining, Faculty of Geoengineering, Mining and Geology, Wrocław University of Science and Technology, 50-370 Wroclaw, Poland

**Keywords:** belt conveyor, condition monitoring, overheated idlers, thermal imaging, image classification, convolutional neural network, binary classification

## Abstract

Mechanical industrial infrastructures in mining sites must be monitored regularly. Conveyor systems are mechanical systems that are commonly used for safe and efficient transportation of bulk goods in mines. Regular inspection of conveyor systems is a challenging task for mining enterprises, as conveyor systems’ lengths can reach tens of kilometers, where several thousand idlers need to be monitored. Considering the harsh environmental conditions that can affect human health, manual inspection of conveyor systems can be extremely difficult. Hence, the authors proposed an automatic robotics-based inspection for condition monitoring of belt conveyor idlers using infrared images, instead of vibrations and acoustic signals that are commonly used for condition monitoring applications. The first step in the whole process is to segment the overheated idlers from the complex background. However, classical image segmentation techniques do not always deliver accurate results in the detection of target in infrared images with complex backgrounds. For improving the quality of captured infrared images, preprocessing stages are introduced. Afterward, an anomaly detection method based on an outlier detection technique is applied to the preprocessed image for the segmentation of hotspots. Due to the presence of different thermal sources in mining sites that can be captured and wrongly identified as overheated idlers, in this research, we address the overheated idler detection process as an image binary classification task. For this reason, a Convolutional Neural Network (CNN) was used for the binary classification of the segmented thermal images. The accuracy of the proposed condition monitoring technique was compared with our previous research. The metrics for the previous methodology reach a precision of 0.4590 and an F1 score of 0.6292. The metrics for the proposed method reach a precision of 0.9740 and an F1 score of 0.9782. The proposed classification method considerably improved our previous results in terms of the true identification of overheated idlers in the presence of complex backgrounds.

## 1. Introduction

The automation and robotization of condition monitoring (CM) processes in the mining sector are inevitable steps for increasing the industrial equipment life cycle and improving the safety of operations. Modern automation methods provide solutions to reduce or totally exclude the presence of humans in hazardous environments. Using mobile robots for CM applications can improve the inspection process, as they apply more sophisticated techniques for identification and localization of faults in industrial infrastructures located in mining sites [1,2,3,4,5,6].

Conveyor systems have been developed throughout the past decades and used as the most common method for transporting raw materials in mining sites. The idlers are responsible for supporting the loaded bulk materials’ weights that are carried by the belt [7,8]. Faulty idlers can cause serious damage in conveyor systems [9,10].

The minimum L10 life requirement (the amount of time in which 10% of the bearings will fail) or, in other words, the lifetime of idler bearing in normal environmental conditions, should be 50,000 h or 5–7 years. However, this time can considerably decrease due to environmental conditions and belt speed [11,12].

Harsh environmental conditions in mining sites can considerably reduce the life span of the conveyor system idlers. Dust, high temperature, and sunlight can be considered as environmental factors that can affect the life span of idlers. The idler sealings are usually poor; therefore, the accumulation of dust in conveyor systems can increase the rotation resistance of the conveyor idlers. Similarly, high ambient temperatures and sun reflection in hot, dry climates can increase the bearing temperature, which reduces the bearing life span [13]. The idler bearings’ condition can be monitored by analyzing their surface temperatures, noise emissions, and vibrations [14,15]. Robotic-based infrared thermography (IRT) can be considered a reliable method for automatic CM of idlers [16,17].

Different fault signatures in rotating machinery can be caused by misalignment, and bearing damage and mass imbalance are recognizable by IRT methods [18,19,20,21]. Analyzing the temperature patterns on the outer surface of idlers can give us information to evaluate the degree of deterioration on inspected idler bearings [22,23]. Detection of anomalous temperature signatures in thermal or infrared (IR) images can help us to determine the overheated idlers [24].

Two different methods can be used to measure the thermal condition of idlers. In quantitative methods, we need to determine the exact temperature of the idler surfaces. On the other hand, in the qualitative methods, the acquired thermal signatures are analyzed to find the characteristic patterns based on relative temperatures of equipment in a captured scene. In our study, we chose to identify the overheated idlers base on qualitative methods; therefore, the relative temperature values of detected hotspots in IR images were analyzed with respect to other areas [25,26,27,28].

Modern IR image analysis software has the capability to prepare an inspection report. However, despite its ease of use and functionality, the manual evaluation of data is time-consuming for inspectors [29]. Furthermore, in our case, for conducting inspection programs in a large mining site, where series of conveyor systems need to be inspected individually, the manual evaluation of reports can become more complicated.

Over-centralized distribution of pixels and their low-intensity contrast are the main features of the IR images that bring difficulties in correct identification of temperature anomalies within the region of interest (ROI). The classical threshold-based segmentation techniques can accurately segment the ROI areas in images that are captured in visual light. However, due to the nature of IR images, the accuracy of final results can be affected by over- and undersegmentation [30,31,32]. Furthermore, in our case study, the presence of reflective objects with high emissivity values that are not related to idlers can create complex background conditions which directly affect the accuracy of segmentation results [33].

In this paper, we developed solutions for improving the proposed CM method in our previous study [24]. Therefore, we focused on proposing a classification method for accurate detection and classification of the overheated idlers in segmented frames from other thermal sources that were not related to idlers. The development of a methodology for the classification of thermal defects in segmented IR images can allow us to improve the number of true detections and the overall accuracy of the proposed CM method. So, the main contributions are:1.A novel method based on binary classification with CNN was developed for classification of overheated idlers in segmented frames.2.We showed that our proposed CM method can accurately classify the overheated idlers, even in IR images with complex backgrounds.3.Our novel CM method based on the deep learning approach was compared against an IRT method that was specifically developed for identification of the overheated idlers in IR images.

## 2. Literature Review

In this section, we firstly focus on the importance of application of robotic-based IRT for the reliability of conveyor systems in mining environments. The length of a conveyor belt in mining sites can reach several kilometers, while hundreds of idlers that support the belt, and the loads need to be monitored by technicians [34,35]. The manual condition monitoring methods are time-consuming and inaccurate due to low frequency of inspections. Furthermore, working in high-pollution environments can be harmful for human health [36,37,38]. By using automatized CM solutions in mining sites, operators can considerably reduce the number of unplanned shutdowns and increase Overall Equipment Efficiency (OEE) [39,40,41].

Autonomous or semi-autonomous robotics IRT are categorized as a non-destructive CM method that can be used as a proper solution for automatic detection of faulty idlers in conveyor systems. In IRT-based techniques, the emitted IR radiation from objects’ surfaces is analyzed for identification of temperature anomalies [42].

IRT methods are used in a wide range of fields, including medical imaging, non-destructive testing, defect detection in civil structures, and so on [43]. However, due to the complexity of mining environments and various factors that need to be considered in developing IRT methods, the application of machine learning methods in automated diagnoses of IR images captured in mining environments is still in the early stages [44,45,46].

Most of the proposed automated diagnostic system for analyzing IR images of industrial infrastructures consist of three steps: estimation of ROI, extraction of relevant features and, finally, classification of extracted features [18,47,48]. The definition of ROI can reduce the complexity of the analyzed images and improve the feature extraction results in the next steps. Afterward, the desirable features should be processed and extracted in such a way that it gives us enough information to understand the current state of the monitored equipment. Finally, in the classification stage, the extracted features from previous stages are studied to understand whether or not the images contain possible signs of thermal defects.

In [16,20,24,49], researchers proposed a different CM method using the IRT method for identification of overheated idlers in conveyor systems located in mining sites. From the investigations with IR images taken with mobile robots, some researchers only focused on the segmentation of hotspots without discriminating against the defected idlers. In [16], for identifying thermal defects in idlers, researchers proposed a method based on Canny edge detection and blob detection techniques. In other researches, alternatively, overheated idlers were segmented by the following methods: color-based thresholding [20], an outlier detection technique [24], and a simple thresholding method based on maximum and minimum temperatures [49]. The mentioned methods have been referred to as classical methods. Classical IRT methods only focus on excluding hotspots from backgrounds, but they do not classify them based on their IR sources.

For the detection of the idlers from their background, different methods based on deep learning are proposed. For instance, a CM method based on object detection, Hough line transform, and template matching algorithms are proposed by [36]. However, the presence of other thermal sources and objects with high emissivity values in complex backgrounds lead to hotspots with atypical geometries that the mentioned method could not address.

The weaknesses of the related works are grouped into three categories: the first category focuses only on the detection of hotspots; the second category uses stationary camera systems for CM of equipment; and the third category handles deep learning techniques but does not propose methods for classification of other thermal sources that can be wrongly identified as thermal defects. Information is presented in more detail in Table 1.

## 3. Materials and Methods

The main idea of the methodology is described in Figure 1. Firstly, the captured data by mobile robot including IR and RGB videos were loaded and prepared for preprocessing. Afterward, the total number of frames from both sources were extracted. Furthermore, to reduce the size and complexity of captured frame, we choose to convert the colored IR frames into 8-bit grayscale ones.

In some frames, due to the existence of different thermal sources that are not related, the segmentation algorithm cannot always precisely exclude hotspots (overheated idlers) due to oversegmentation. For improving the oversegmentation issue, we choose to preprocess the ROIs using different preprocessing techniques. In this study, intensity normalization, Contrast Limited Adaptive Histogram Equalization (CLAHE), and the Fast Non-Local Means Algorithm (FNLM) were used.

For segmenting overheated idlers in preprocessed frames, an outlier detection method based on the interquartile range (IQR) technique was used to calculate the optimal threshold value in each frame.

Based on the visual analysis of the segmented frames, we noticed that in some frames other thermal sources including sun reflection or other overheated equipment were wrongly segmented and recognized as overheated idlers. This can reduce the accuracy of the CM method. For the identification of hotspots in segmented frames, we proposed a classification method based on binary classification technique using a CNN. The different stages of the proposed methodology were developed in the Python programming language using OpenCV library for image processing and did not have major changes. Furthermore, the Keras and TensorFlow libraries were used for developing the proposed image classifier.

### 3.1. Region of Interest Estimation

Through the conducted experiments, the inspection mobile robot moved alongside the conveyor systems and captured different data, including IR and RGB videos. The mobile robot camera system was installed on a fixed hand above the robot and pointed toward the conveyor system. The mobile robot followed a straight line with constant speed; therefore, we did not have major changes in the camera system POV toward the conveyor system. The captured RGB and IR videos from the conveyor systems contain information on target pixels that are related to idlers and redundant areas. For improving the segmentation accuracy, we excluded the non-ROI areas from the original captured frames.

Through analyzing the extracted frames, we understand that idlers that are located on the right side of the conveyor (from mobile robot POV) were not always visible for being analyzed in extracted IR and RGB frames; therefore, for improving idler detection, we chose to define ROI on extracted frames for capturing idlers that were on the left side of the conveyors.

By observing neglectable changes in the camera system POV through the sequence of extracted frames, we could accurately predict the location of idlers. Different detection algorithms can be used for following the idlers in the sequence of the frame. In our work, we chose to consider a rectangular ROI with predetermined size: 200 × 200 pixels with a fixed position to extract ROIs from captured IR and RGB frames. The size of the extracted ROIs covers the idler sections, while it is considerably smaller than the originally captured frames Figure 2.

### 3.2. Pre-Processing of Region of Interests

The captured IR images are characterized by noise due to illumination and contrast variation; therefore, the use of automatic segmentation methods for detection of overheated idlers can become a challenging task. To address this issue, in this paper, intensity normalization, CLAHE, and FNLM were used for improving the overall quality of extracted IR images.

#### 3.2.1. Normalization

The intensity value of each pixel in an original frame is predefined by the IR camera, considering the hottest and coldest point within the frame borders. For defining statistical parameters that work well for analyzing ROIs, we need to recalculate the pixels’ values with respect to the pixel distribution in defined ROIs. Normalization of the ROIs can reduce the seasonal differences and lets us define statistical parameters that work in ROIs with varying temperature ranges. The normalization was performed using the following equation:(1)$$N_{i}=\frac{TS_{i}-TS_{\min}}{TS_{\max}-TS_{\min}}$$

In Equation (Equation 1), $N_{i}$ can be defined as the normalized value of pixel *i*, where $TS_{i}$ is defined as the intensity value of pixel *i*. Furthermore, the maximum and minimum values of pixel intensities within the ROIs are defined as $TS_{max}$ and $TS_{min}$.

#### 3.2.2. CLAHE Method

Histogram equalization (HE) is an elementary method to improve the contrast of underexposed and overexposed images. However, in the HE method, changes in natural brightness of the processed images can cause undesirable noise in the results [50]. Adaptive Histogram Equalization (AHE) is the improved version of the HE method [51]. In AHE, the input image is split into smaller areas, while the cumulative distributive function (CDF) is generated for each of these smaller images. In this method, the noise can considerably increase when the image histogram slope is steep. For addressing this issue, the CLAHE method is proposed. CLAHE is an improved version of the AHE that works in the same way. In the CLAHE method, the extracted histogram of the input image is clipped at specific values for limiting the amplification before computing the CDF, which can considerably reduce the amount of the unwanted noise in the final results. The calculation of CLAHE is performed as follows:(2)$$p=\left(p_{\max}-p_{\min}\right)*P\left(f\right)+p_{\min}$$

In Equation (Equation 2), the maximum and minimum intensity value of pixels in an image are defined by $p_{\max}$ and $p_{\min}$, while P(f) defines the cumulative probability distribution function, and *p* represents the assigned value to each pixel after applying CLAHE.

#### 3.2.3. Fast Non-Local Means Algorithm

The Non-Local Means Denoising Algorithm (NLM) is an effective method for reducing the unwanted noise in images Figure 3. As opposed to Gaussian, median, and Wiener filters, the NLM method uses the Euclidean distance as a weight; therefore, it can provide better results. The NLM denoising algorithm can be defined as follows:(3)$$NL\left[I\right]\left(m\right)=\sum_{N\in I}\omega \left(m,n\right)I\left(n\right)$$

In Equation (Equation 3), the weight: ω(m,n) can be expanded as follows:(4)$$\omega \left(m,n\right)=\frac{1}{Z(m)}\sum e^{-\frac{G_{\sigma }\left(\tau \right){∥I\left(M+\tau \right)-I\left(n+\tau \right)∥}_{2}^{2}}{d^{2}}}$$
where τ describes the number of pixels in an image and $G_{\sigma }\left(\tau \right)$ can be defined as Gaussian distribution of pixels with size $\sigma^{2}$ of the number of available pixels in the background. Furthermore, ${∥I\left(M+\tau \right)-I\left(n+\tau \right)∥}_{2}^{2}$ describes the differences in intensities between adjacent pixels based on calculating the euclidean distance values. The leveling constant is set by *Z*(*m*):(5)$$Z\left(m\right)=\sum_{n}e^{-\frac{G_{\sigma }\left(\tau \right){∥I\left(m+\tau \right)-I\left(n+\tau \right)∥}_{2}^{2}}{d^{2}}}$$

The calculation of ω(m,n) in FNLM is modified from one dimension to two dimensions. The modified version of ω(m,n) can be defined as follows:(6)$$\omega \left(m,n\right)=\frac{1}{Z(m)}H_{i}\left(I\left(m+s\right)-I\left(m-s\right)\right)$$

In Equation (Equation 6), τ is defined based on n−m, *s* is defined as m+τ, and $H_{i}$ can be describe as follows:(7)$$H_{i}\left(s\right)=\sum_{q=0}^{s}e^{-\frac{{∥I\left(q\right)-I\left(q+\tau \right)∥}_{2}^{2}}{d^{2}}}.$$

By simplifying the process through one-dimensional computations, the FNLM method can denoise the image four times faster than the NLM method [52,53,54,55,56,57,58,59,60].

### 3.3. IR Image Segmentation

Image segmentation is a process of dividing images into regions that may be meaningful for extracting the desirable objects or constituent areas. Thresholding techniques are considered as the simplest way of performing image segmentation in IR images. Generally, in IR images, regions with high temperatures are correlated to the foreground; therefore, pixels with high intensity represent the heated objects. In the proposed method, the optimal threshold value for captured IR images can be defined based on analyzing the statistical features that are extracted from the IR images’ histogram.

#### 3.3.1. Histogram Analysis of IR Images

An IR image histogram describes the tonal or color distribution of pixels. The tonal distribution of pixels in grayscaled IR images refers to discrete temperature values; therefore, the analysis of IR images’ histogram from a statistical point of view is a useful approach for detecting temperature anomalies in monitored equipment. Different statistical features such as: mean value, variance, and standard deviation can be extracted to describe the pixel tonal distribution in IR images. For a gray-leveled IR image, the first-order histogram probability P(g) is computed as follows [61]:(8)$$P\left(g\right)=\frac{L(g)}{M}$$

In Equation (Equation 8), L(g) is the number of gray levels *g*, where the total number of pixels in a processed image is defined by *M*. In gray-scaled IR images, the total number of available levels for the pixel *L* span into [0,256]. The general brightness, or in other words average temperature, in a captured frame can be defined by mean value as follows:(9)$$\bar{g}=\sum_{g=0}^{L-1}g\;\cdot \;P\left(g\right)$$

Furthermore, the dispersion of a set of data points around their mean can be defined by variance value as follows:(10)$$\sigma_{g}^{2}=\sum_{g=0}^{L-1}{\left(g-\bar{g}\right)}^{2}\;\cdot \;P\left(g\right)$$

The standard deviation or the square root of the variance indicates the image contrast. It can be considered as an important factor for identifying the thermal anomaly, as analyzing temperature distribution in IR images is a key index of detection of possible defects.

#### 3.3.2. Anomaly Detection

In this work, a qualitative-based method is proposed for identification of hotspots in preprocessed frames. In IR images with uniform background, the defected idlers can be recognized as hotspots. For separation of the hotspots from the background, thermal anomalies in idlers’ surfaces are treated as outliers. Outlier detection can be described as a problem of finding data that cannot be defined in the range of normal behavior. The interquartile range (*IQR*) can be used to extract histogram features that can help us to define the outliers.

The *IQR* defines the difference between the first and the third quartile: IQR=Q3−Q1 where Q1 and Q3 can be calculated as follows [62,63]:
(11)$$\begin{array}{cc} \; & Q1=\bar{g}-0.675\sigma \\ \; & Q3=\bar{g}+0.675\sigma \end{array}$$

The outliers can be extracted and classified in two different groups, namely: mild and extreme outliers. The lower and upper bounds can be calculated as Q1−3IQR and Q3+3IQR. The values that are either among inner or outer bounds are classified as mild outliers; on the other hand, extreme outliers can be considered as values that are beyond outer bounds [64,65].

The optimal thresholder value can be defined as *T* in each frame, while α and β refer to the segmented frames and preprocessed ROIs, respectively. Furthermore, *W* and *H* can be considered as the processed ROI height and width, while *x* and *y* represent the pixel coordination.
(12)$$\begin{array}{cc} \; & \beta \left(x,y\right)=\left\{\begin{array}{cc} 1 & \;\mathrm{if}\;\alpha (x,y)>T\\ 0 & \;\mathrm{if}\;\alpha (x,y)\leq T \end{array}\right.\\ \; & \forall 0\leq x<W,0\leq y<H \end{array}$$

To separate the hotspots from background in α, the pixel values were set to 1 when they were greater than computed T, otherwise they were set to 0:(13)$$\begin{array}{cc} \; & \gamma \left(x,y\right)=\left\{\begin{array}{cc} \alpha (x,y) & \;\mathrm{if}\;\beta (x,y)=1\\ 0 & \;\mathrm{if}\;\beta (x,y)=0 \end{array}\right.\\ \; & \forall 0\leq x<W,0\leq y<H \end{array}$$

The pixels that are related to overheated idlers cannot always be defined as mild outliers. Furthermore, the definition of overheated idlers based on mild outliers can cause oversegmentation. For addressing the oversegmentation issue, we choose to extract extreme outlier values in extracted IR frames and consider them as the optimal threshold values for segmentation of possible hotspots, as shown in Figure 4.

### 3.4. Convolutional Neural Networks

During the visual inspection of the results, we found out that the proposed segmentation method could accurately segment the overheated idlers from the background in the majority of extracted frames. However, due to overestimation in some frames, other thermal sources were wrongly segmented as overheated idlers.

The objective of this section is to propose a method to accurately classify the segmentation results in two different classes considering the differences between hotpot shapes in segmented frames. The first class consists of frames where the segmentation method could accurately separate overheated idlers from the background, while the other class consists of hotspots that were not related to idlers.

A CNN is an ANN (artificial neural network) with different numbers of intermediate layers. The CNN models are able to automatically detect the specific features on each image and then train a classifier model on them [66,67]. CNNs performance in classification tasks can be improved when the size of the given dataset continues to increase. A developed CNN can learn to extract specific features from a given dataset and classify them at the same time.

Deep learning models that use CNN architecture are suitable for image processing classification problems. In CNNs, different layers in a network work as a detection filter for the recognition of specific patterns or features. The first layer of a CNN model can detect major features in an image that can be easily interpreted, while in middle layers, more abstract features can be detected. Finally, in the last layer, the collected details from previous layers are accurately merged and classified [68].

#### 3.4.1. Data Collection and Description

Through the experiments, two conveyor systems that were located in an opencast mining site were inspected. The inspected conveyor systems were used for the continuous transportation of mineral materials from the mine pit to the bunker. The length of the investigated conveyor systems reached one hundred meters and is designed to operate in harsh environmental conditions. The opencast mining site in this study is located in Jaroszów, 50 km to the west of Wrocław. The length of the inspected parts of the conveyor systems were 150 m, where there was a 1.45 m space between each idler. Furthermore, the idlers’ diameter was 1.33 m and the belt length was 0.8 m.

A mobile inspection robot was used for data accusation Table 2. The inspection mobile robot is specially designed for Wrocław University of Science and Technology as a mobile platform that can conduct inspection missions in harsh environmental conditions, as shown in Figure 5.

The precision of an IRT method is directly related to environmental parameters and the IR camera specifications Table 3. The parameters such as environmental conditions and emissivity values of the objects should be taken into account for accurate identification of overheated equipment [33]. In our case study, one can notice that in a raw IR image there are different radiation sources that can be wrongly identified as overheated idlers Figure 6.

In opencast mining sites, solar radiation can warm up the conveyor system surfaces, especially modules that can absorb the considerable amount of sun energy. Our experiment was conducted on a sunny day. As long as we organized the measurement session a few weeks before the experiment, it was almost impossible to predict the weather. However, in the proposed method, we tried to provide a solution that can be applicable in different environmental conditions. Furthermore, in our case, the solar radiation was mostly blocked by ceiling; therefore, their affect on the idler surface temperature was negligible.

Examples presented here show the application of IRT in industrial data sets is always challenging. As a result, the classical image segmentation techniques cannot provide accurate detection results due to instances of different thermal sources in mining areas, as shown in Figure 7. prances of different thermal

#### 3.4.2. Binary Classification

In this work, we developed a binary classification model for classification of the segmented hotspots to identify frames where overheated idlers are correctly segmented Figure 8.

The proposed classification model includes three blocks, where each block is composed of a convolution layer with a 3 × 3 filter, followed by a maximum grouping layer. The ReLU activation function was used in each conventional layer. The depths of our architecture for the three blocks are 32, 32, and 64 respectively. A value of zero (0) or one (1) should be predicted for each input image, as we used a binary classification. The sigmoid type of activation was used to regularize the deep neural network at the final output layer. Regularization simulates the different possible changes on the structure of the network and increases the strength of the nodes in the developed network. Finally, for compiling the data, the binary cross-entropy method was used as a loss function, and the Adam method was used as the optimizer to form a classifier on the image data sets Figure 9.

### 3.5. Training and Testing the CNN

Three different data sets, which consist of segmented IR images captured from two different conveyor systems, were used for training the CNN model. Through the experiments, the inspection robot moved alongside conveyor system number one (back and forth) and conveyor system number two (forth). As a result, three different data sets, 6275 frames from conveyor one (moving forward), 6135 frames from conveyor 2 (moving backward), and 10896 frames from conveyor two (moving forward), were selected and underwent preprocessing stages and segmentation. Afterward, the segmented frames were filtered based on the size of the segmented hotspots; therefore, in each segmented frame, the number of non-black pixels was calculated, and frames that the number of non-black pixels was above 400 pixels were kept for further analysis.

The classifier was trained using images labeled by experts. We labeled each segmented image as “contained an overheated idler” or “other thermal sources”, as shown in Figure 10. Furthermore, the filtered and labeled frames from the mentioned data sets are combined together for training the CNN model. A total of 1216 frames from the three different data sets was selected for the training process. The images are randomly split, and 20% of the total images were used for testing images and the remainder as training images.

## 4. Evaluation

For evaluation of the proposed classifier, we computed the following performance metrics: sensitivity, precision, accuracy, and F1-score.
(14)$$\;\mathrm{Accuracy}\;=\frac{\left(TP+TN\right)}{\left(TP+FN)+(FP+TN\right)}$$
(15)$$\;\mathrm{Sensitivity}\;=\frac{\left(TP\right)}{\left(TP+FN\right)}$$
(16)$$\;\mathrm{Precision}\;=\frac{\left(TP\right)}{\left(TP+FP\right)}$$
(17)$$\;F1\;\mathrm{Score}\;=\frac{\left(2*Precision*Sensitivity\right)}{\left(Precision+Sensitivity\right)}$$

Here, accuracy is the proportion of correctly classified overheated idlers among the whole population. Sensitivity is measured as the proportion of true positive cases that are correctly predicted by the classifier, while specificity is the prediction of true negative cases that are correctly predicted. Precision is the proportion of the correct predictions in the confusion matrix out of all positive predictions. Furthermore, the F1-score is the harmonic mean of precision and sensitivity. The coefficient takes into account the *TP* (true positive), *TN* (true negative), *FP* (false positive), and *FN* (false negative) factors for scoring the model. The ideal value of these metrics is 1 and is the target for the models in this study.

## 5. Results

In this section, we evaluate the performance of the proposed method in the detection of overheated idlers. Furthermore, the performance of the proposed method is compared with our previous research using test data randomly chosen from 20% of the whole data set [24].

The confusion matrix is used to display the results of classifications and generate all of the performance metrics. Our binary classification model could accurately classify the segmentation results in over 97% of studied cases, as shown in Figure 11.

The training and validation loss graph of the trained model is shown in Figure 12. The loss of validation data significantly decreased until the 6th epoch, but on the next epoch, the trends tended to increase. This means that our model converged fast until the 6th epoch, and the greater epoch would not give any significant change on the classification result.

The training and validation accuracy curves are shown in Figure 13. The curves indicate that through each training epoch the accuracy parameter varies and then continuously converges to a certain accuracy level with a lot of small oscillation. The trends also show that the classifier did not affect by significant overfitting or underfitting.

The receiver operating characteristic curve (ROC) for a binary classification model is shown in Figure 14. ROC is a receiver operating curve, which is one of the model’s measuring metrics. The area under curve can be defined as the AUC that indicates the performance of the model. It is a graph that shows how the true positive rate (TPR) and the false positive rate (FPR) are related. For the studied binary classification model, the ROC was 100%.

Table 4 compares the performance metrics of the proposed method and our previous work in the identification of overheated idlers in captured IR frames. The precision value shows that the binary classification method has few errors. In this table, it is shown how the shape-based classification of segmented hotspots affects the accuracy metrics. It is demonstrated that the false identification of other thermal sources as overheated idlers in our previous research had a major effect on the performance of the CM method. Reducing the portion of false negatives by more than 54% led to an increase in the F1 score value from 0.62 to 0.97 in this work.

## 6. Conclusions

In this paper, we solve the problems related to the identification of overheated idlers in IR images with complex backgrounds. We demonstrate that the identification of overheated idlers in IR images captured in real case scenarios is a challenging task. For the proposed methodology, the following points were covered: Firstly, in Section 3.2.2, we replaced our previous noise reduction method with the FNLM method, which could help us to reduce the amount of presented noise more efficiently. Secondly, the proposed binary classification methodology allows the detection of frames that other thermal sources wrongly segmented as overheated idlers.

The value of the accuracy metrics in Table 4 for the proposed method is greater than 0.9795, which implies that the discussed binary classification method is reliable, confirmed by the precision value being greater than 0.9740.

Future work assumes the improvement of the CM method in terms of adaptability to different conveyor systems that are located in different mining sites. Furthermore, our team is interested in working on methods that can be used for the detection and localization of overheated idlers in global navigation satellite system denied environments, such as deep underground mines, by fusion of LIDAR and IR images captured by an inspection mobile robot.

## Figures and Tables

**Figure 1 sensors-22-10004-f001:**
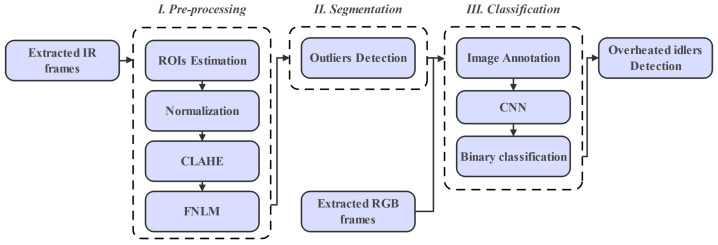
The proposed procedure flowchart.

**Figure 2 sensors-22-10004-f002:**
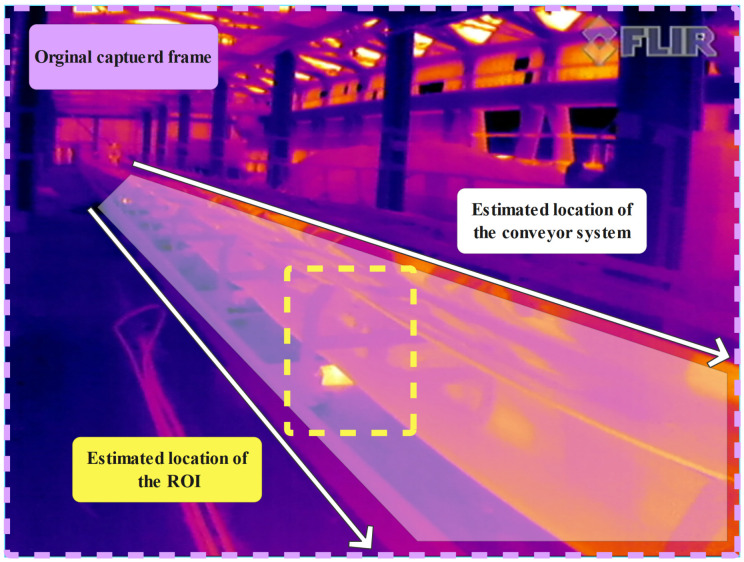
The location of a predefined ROI on an original frame.

**Figure 3 sensors-22-10004-f003:**
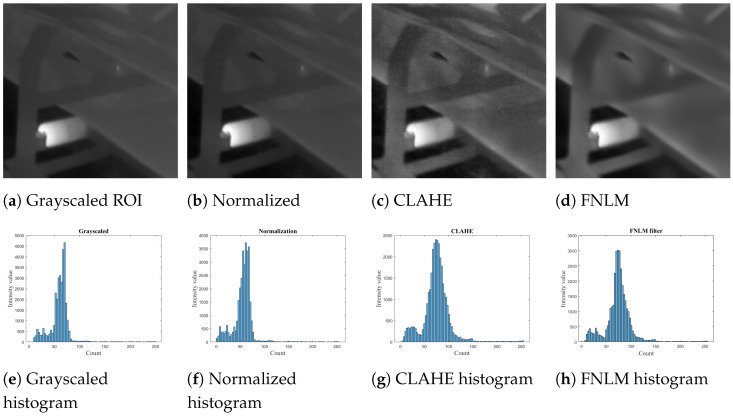
Comparison of a preprocessed and an original IR image after modifications through preprocessing stages.

**Figure 4 sensors-22-10004-f004:**
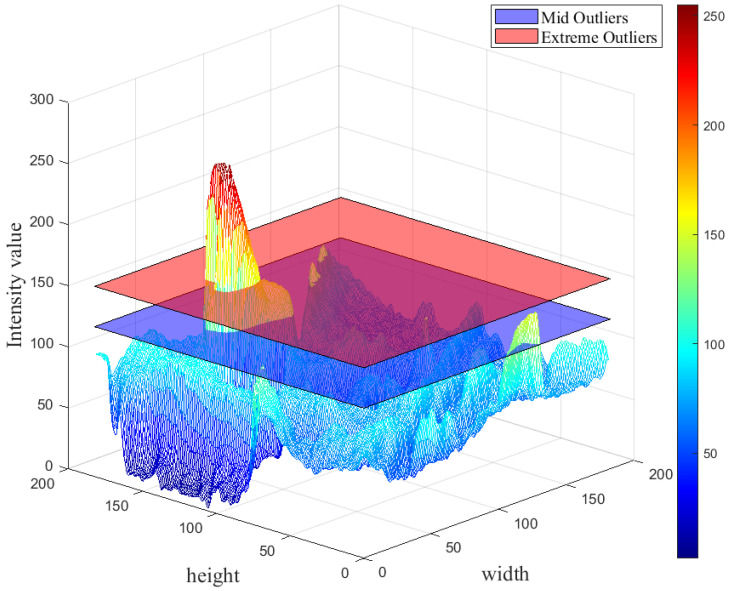
Comparison of detected outliers by mid and extreme values in a preprocessed ROI.

**Figure 5 sensors-22-10004-f005:**
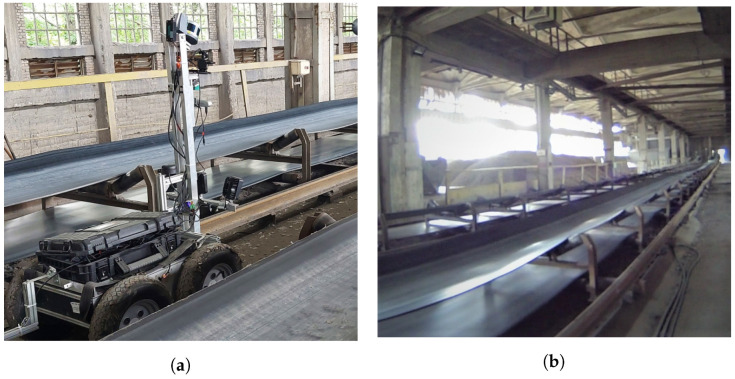
(**a**) Mobile robot during inspection. (**b**) A general picture of the mining site.

**Figure 6 sensors-22-10004-f006:**
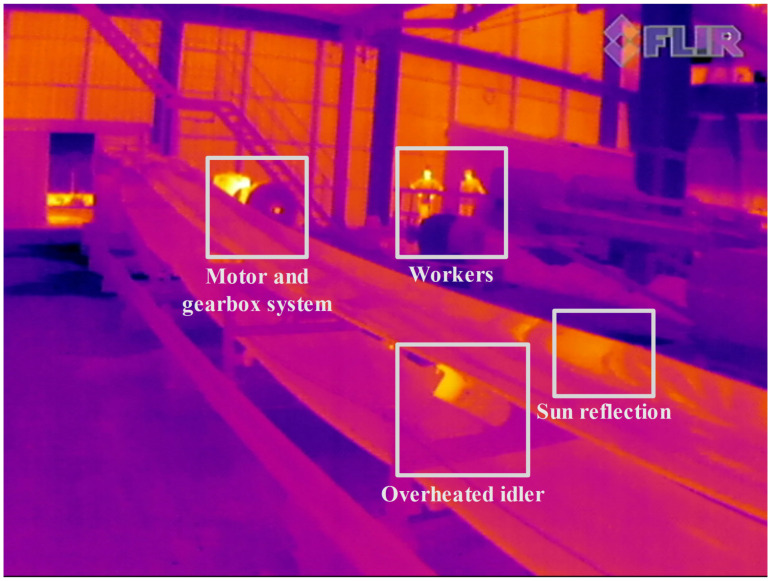
Examples of IR sources that are not related to idlers.

**Figure 7 sensors-22-10004-f007:**
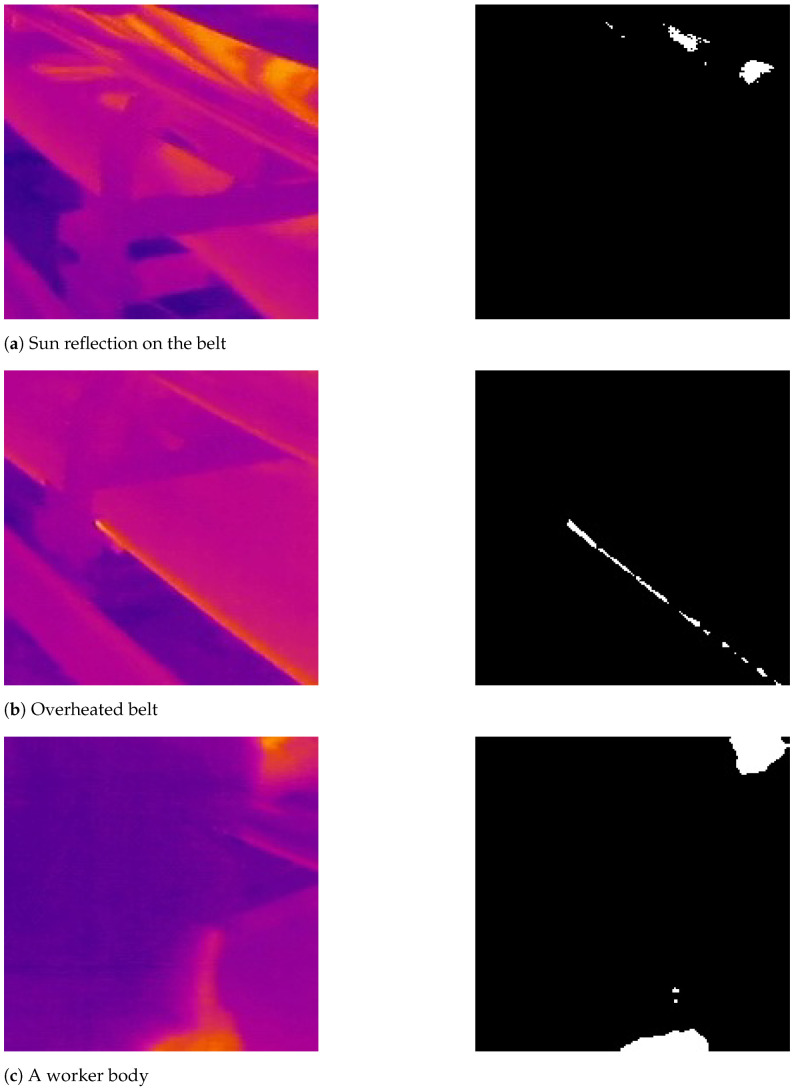
Segmentation results of overheated objects that were not related to idlers.

**Figure 8 sensors-22-10004-f008:**
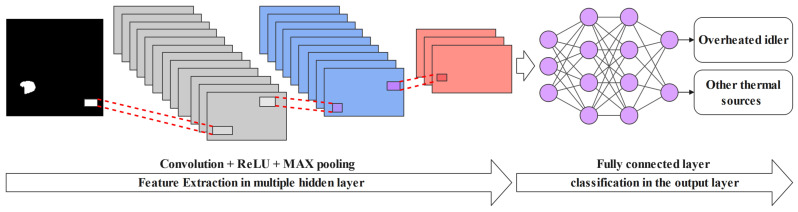
Simplified flowcharts of binary classification procedure.

**Figure 9 sensors-22-10004-f009:**
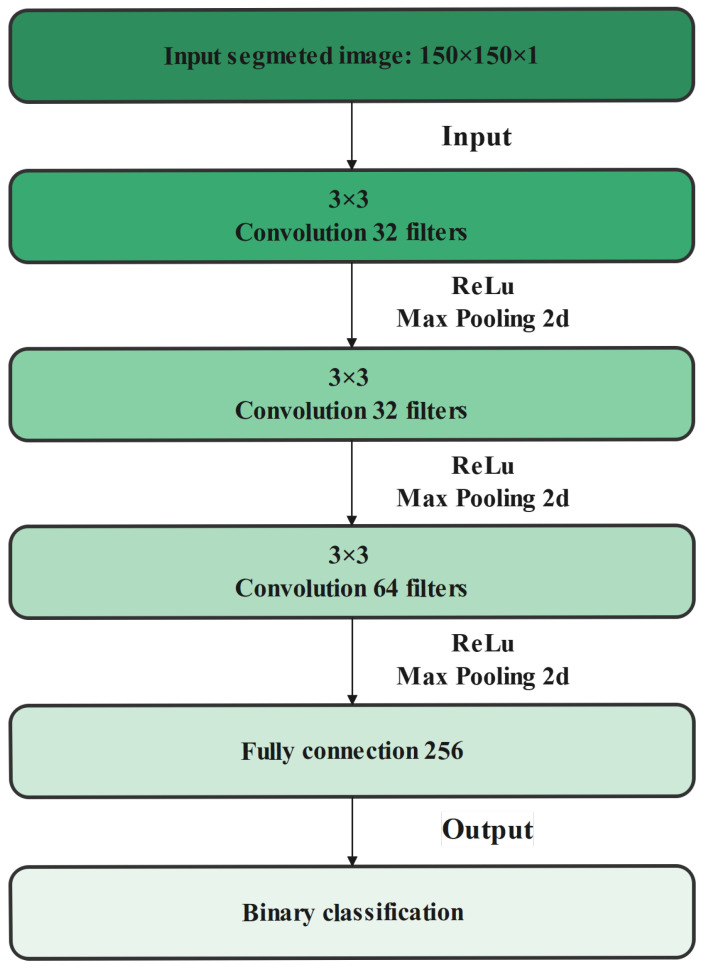
Summary of the proposed CNN architecture.

**Figure 10 sensors-22-10004-f010:**
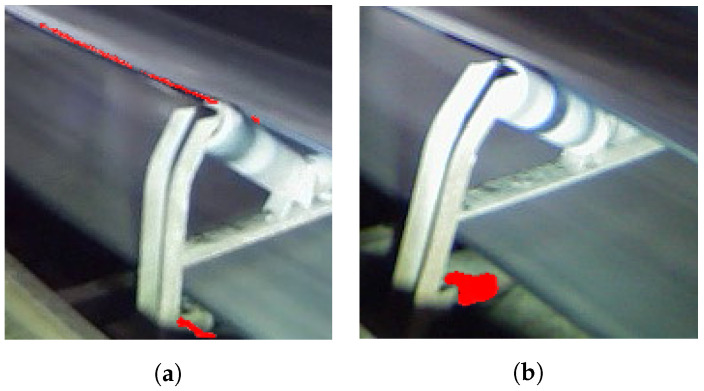
Image annotation process base on fusion of IR and RGB images. (**a**) Segmentation of other thermal sources; (**b**) segmentation of an overheated idler.

**Figure 11 sensors-22-10004-f011:**
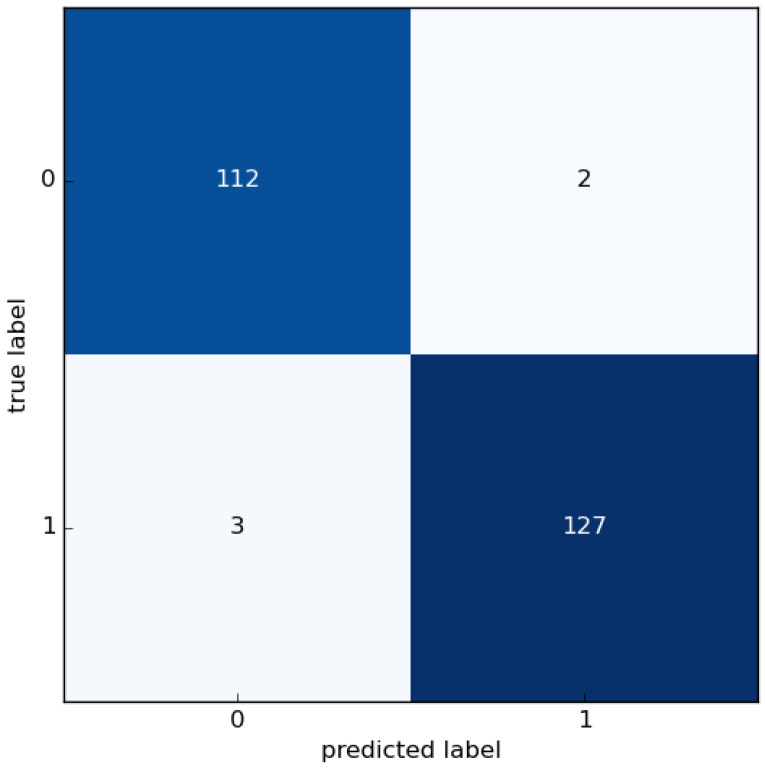
Confusion matrix of the binary classification.

**Figure 12 sensors-22-10004-f012:**
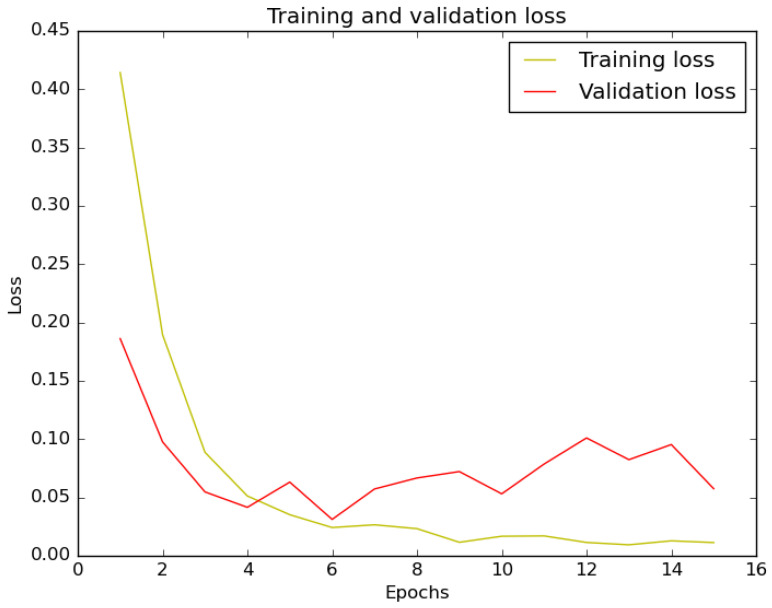
Training and validation loss plot for the binary classification model.

**Figure 13 sensors-22-10004-f013:**
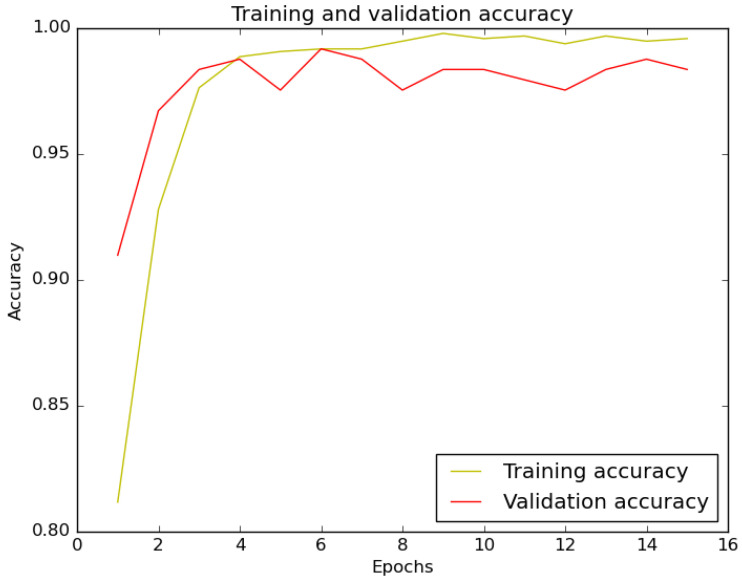
Accuracy plot for the binary classification model.

**Figure 14 sensors-22-10004-f014:**
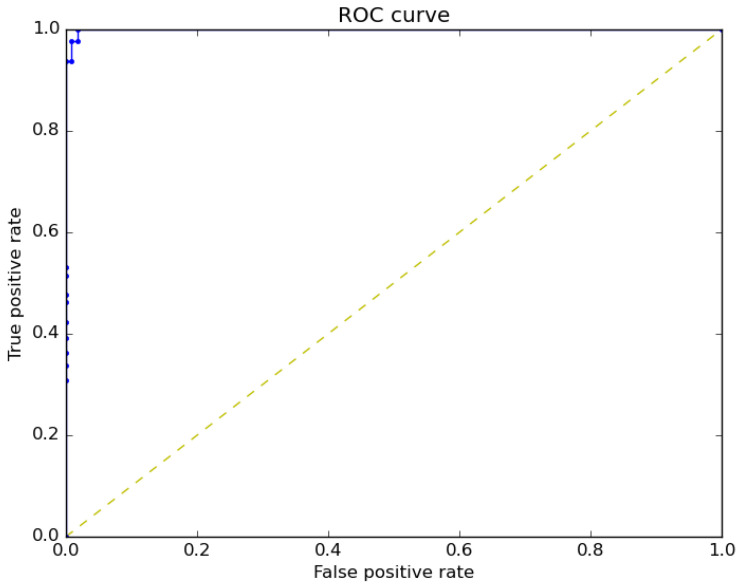
ROC Curves of the binary classification model.

**Table 1 sensors-22-10004-t001:** Comparison between research that proposed different IRT methods for identifying overheated idlers in conveyor systems.

Methods	Descriptions	Weaknesses
Dabek et al. [16]	Identifies hotspots in captured IR videos based on Canny edge and blob detection methods.	Lacks a method to classify segmented hotspots. It does not propose a solution for frames with complex backgrounds.
Siami et al. [24]	Proposed an image processing pipelines for identification of hotspots based on outlier detection method.	It does not present a solution for classification of segmented hotspots based on their IR sources.
Szurgacz et al. [49]	Proposed a CM method based on simple thresholding technique.	Tested on limited number of IR images. IR images were captured manually by inspectors.
Liu et al. [36]	Proposed a method for identification hotspots in IR images based on Hough line transform and template matching algorithm.	Tested on limited number of IR images on a controlled environment. The data were captured by stationary camera systems.

**Table 2 sensors-22-10004-t002:** The main characteristics of the mobile inspection robot.

Locomotion type	Wheeled, skid steering
Navigation systems	Autonomous (internal computer)
	Manual (pilot using remote computer connection)
Internal operating software	Robot Operating System (ROS)
Power system	Internal battery, 24 V
Max gross weight	140 kg
Maximum payload capacity	75 kg

**Table 3 sensors-22-10004-t003:** The IR camera system specification.

Parameter	Value
Resolution	640 × 480 pixles
Frames per second	25 fps
Observation angle	45${}^{\circ }$
Mounting height	100 cm above shelf

**Table 4 sensors-22-10004-t004:** Comparison of the precision, recall, and F1 score metrics for overheated idler detection in IR images using binary classification and Siami et al.’s work [24].

Measures	Siami et al.	Siami et al. + FNLM + Binary Classification	Improvement
True Positive	112	112	
False Positive	132	3	
False Negative	0	2	
Sensitivity	1	0.9825	−1.75%
Precision	0.4590	0.9740	51.50%
Accuracy	0.4590	0.9795	52.05%
F1 Score	0.6292	0.9782	34.90%

## Data Availability

Archived data sets cannot be accessed publicly according to the NDA agreement signed by the authors.

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
