# Peer review of "Automated Identification of Overheated Belt Conveyor Idlers in Thermal Images with Complex Backgrounds Using Binary Classification with CNN"

_sensors, 2022, doi:10.3390/s222410004_

Round 1

Reviewer 1 Report

In this work, the authors proposed an automatic robotics-based inspection for condition monitoring of belt conveyor idlers using infrared images. The paper gives a well explanation of the above work. However, it needs to be improved based on provided comments.  

·         Please mention models, versions, and company producer of the equipment and software packages used in this study.

·         Since the authors mentioned the equipment life cycle, do they consider the life cycle of the equipment in a harsh environment in their results and data?

·         How many times the test is performed since the belt of the conveyor is affected differently each time? Could the authors please specify the error and changes in the results in a Figure or Table?

·         Please proofread the manuscript to tighten minor errors in English grammar.

Author Response

Response to Reviewer 1

Comments and Suggestions for Authors

In this work, the authors proposed an automatic robotics-based inspection for condition monitoring of belt conveyor idlers using infrared images. The paper gives a well explanation of the above work. However, it needs to be improved based on provided comments. 

  1. Please mention models, versions, and company producer of the equipment and software packages used in this study.

Reply:  Thanks for your comment. We introduced the main characteristics of the inspection robot and the main characteristics of the IR camera on Tables 2 and 3. “The robot is custom built for the WrocÅ‚aw University of Science and Technology as an universal mobile platform for inspections”.

Furthermore, we discussed the software packages on the materials and methods section . “The different stages of proposed methodology were developed in the Python programming language using OpenCV library for image processing tasks. Furthermore, Keras and TensorFlow libraries were used for developing the proposed image classifier.”

  1. Since the authors mentioned the equipment life cycle, do they consider the life cycle of the equipment in a harsh environment in their results and data?

Reply: Thank you very much for pointing this out. We add more details to clarify the minimum time that can be considered as standard lifespan of idlers in normal condition and different environmental factors that can affect the idlers life span:

“The minimum L10 life requirement (the amount of time in which 10% of the bearings will fail) or in other words, the quality of idler bearing in normal environmental conditions should be 50,000 hours or 5-7 years. However, this time can considerably reduce in Extreme environments, belt speed.

Harsh environmental conditions in mining sites can considerably reduce the lifespan of the conveyor system idlers. Dust, high temperature, and sunlight can be considered as environmental factors that can affect the idlers' life span. The idler sealings are usually poor therefore, the accumulation of dust in conveyor systems can increase the rotation resistance of the conveyor idlers. Similarly, high ambient temperatures and sun reflection in hot, dry climates can increase the bearing temperature which reduces the bearing life span.”

  1. How many times the test is performed since the belt of the conveyor is affected differently each time? Could the authors please specify the error and changes in the results in a Figure or Table?

Reply: Thank you for your nice reminder. We have added more information on the “Training and testing the CNN” section. Furthermore, we add more details on Table 4 for indicating changes in performance metrics of the proposed mythology and our previous study.

Three different data sets which consist of segmented IR images captured from two different conveyor systems have been used for training the CNN model. Through the experiments, the inspection robot moved alongside conveyor system number one (back and forth) and conveyor system number two (forth). As a result, three different data sets: 6275 frames from conveyor one (moving forward), 6135 frames from conveyor 2 (moving backward), and 10896 frames from conveyor two (moving forward) were selected} and undergo preprocessing stages and segmentation.

  1. Please proofread the manuscript to tighten minor errors in English grammar.

We went through the entire manuscript to eliminate grammatical mistakes.

Reviewer 2 Report

This paper presents a based on infrared image of Belt Conveyor Idlers overheating fault identification method, the main innovations include: a hot spot identification method of image processing under complex background is provided, image segmentation, convolutional neural networks is given, the results have a certain application value. For the author modified reference, the problem as following,:

1,The Belt Conveyor Condition Monitoring method based on the track Mobile robot has been applied more, and the track Mobile robot carry on inspection instruments, such as infrared imager.  Therefore this paper methods of innovation to improve

2,This paper will only some of the more sophisticated image processing methods, such as image segmentation, deep learning methods, convolution neural network is applied to the infrared image processing, not for the algorithm to do deep research and improvement. Therefore this paper deep learning approach innovative general;

3,In figure 5, mobile inspection robot at the side of the belt conveyor for checking, the Idlers overheating fault  of the other side of the Belt Conveyor can be detected?

Author Response

Response to Reviewer 2

Comments and Suggestions for Authors

This paper presents a based on infrared image of Belt Conveyor Idlers overheating fault identification method, the main innovations include: a hot spot identification method of image processing under complex background is provided, image segmentation, convolutional neural networks is given, the results have a certain application value. For the author modified reference, the problem as following,:

  1. The Belt Conveyor Condition Monitoring method based on the track Mobile robot has been applied more, and the track Mobile robot carry on inspection instruments, such as infrared imager. Therefore this paper methods of innovation to improve

Reply: Thanks for your comment. We have improved the “Training and testing the CNN” section with requested information:

“Three different data sets which consist of segmented IR images captured from two different conveyor systems have been used for training the CNN model. Through the experiments, the inspection robot moved alongside conveyor system number one (back and forth) and conveyor system number two (forth). As a result, three different data sets: 6275 frames from conveyor one (moving forward), 6135 frames from conveyor 2 (moving backward), and 10896 frames from conveyor two (moving forward) were selected} and undergo preprocessing stages and segmentation. Afterward, the segmented frames were filtered based on the size of the segmented hotspots, therefore in each segmented frame the number of non-black pixels were calculated and frames that number of non-black pixels were above 400 pixels were kept for further analysis.”

As we discussed in this research, for improving the reliability of the results and demonstrating the performance of the proposed methodology we have improved our data sets by including data that were captured from a different conveyor. In the results section we showed that our robotic based data inspection method could give us enough information for identification of the overheated idles in captured IR and RGB images.  

  1. This paper will only some of the more sophisticated image processing methods, such as image segmentation, deep learning methods, convolution neural network is applied to the infrared image processing, not for the algorithm to do deep research and improvement. Therefore this paper deep learning approach innovative general;

Thank you very much for the comment. We have improved the “Data collection and description” and “Introduction” with new information for indicating the innovative parts of our researches.

As we indicated, identification of overheated idlers in thermal images that captured in real mining sites is a challenging task. We address the overheated idler detection in complex background as an image binary classification task. Binary classification with convolution neural has been discussed in different papers for classification of RGB images however in this research we indicate that the mentioned method can be applied on IR images for classification of the hotspots as well.

  1. In figure 5, mobile inspection robot at the side of the belt conveyor for checking, the Idlers overheating fault of the other side of the Belt Conveyor can be detected?

Thanks for your nice reminder. We have improved the “Region of interest estimation” section with requested information:

“Through analyzing the extracted frames we understand that idlers that are located on right side of the conveyor (from mobile robot POV) were not always visible for being analyzed in extracted IR and RGB frames therefore, for improving the idler detection we chose to define ROI on extracted frames for capturing idlers that were on another left side of the conveyors”

Reviewer 3 Report

In this study, the authors propose several solutions to further advance the condition monitoring (CM) method that they suggested in their previous studies. In this context, they proposed a classification method for the correct detection and classification of overheated rollers in fragmented frames from other thermal sources not related to the idlers. In doing so, they claim to have solved the problem of identifying overheated reels in IR images with complex backgrounds. The CM process proposed in the study for the classification of overheated idlers is innovative compared to Classical IRT methods, and the results obtained are sufficient. It is appropriate to publish the relevant work in Sensors. Consequently, it is appropriate to publish the relevant work in Sensors. The following minor edits should be considered prior to publication.

·       Some typos should be corrected. For example, "sued" on page 13 line 334.

·       All symbols in the manuscript should be reviewed. Symbols used in the text should be shown in the same way as those used in equations. For example, "P(g)" on page 8 line 235, "L" on line 238, "T" on line 265.

·       Figure representations should be the same throughout the manuscript. For example, "Fig11" on page 14 line 366.

·       Reference notations in the text should be arranged. For example, the reference notation in the sentence on page 11 line 306 is "The parameters such as environmental conditions and emissivity values of the objects should be taken into account for accurate identification of overheated equipment.[32]."

·       Some minor visual errors in the figures should be corrected. For example, Figure 9 should be rearranged (output text in frame).

·       The phrase "10" on page 14 line 344 should be revised.

Author Response

Response to Reviewer 3

Comments and Suggestions for Authors

In this study, the authors propose several solutions to further advance the condition monitoring (CM) method that they suggested in their previous studies. In this context, they proposed a classification method for the correct detection and classification of overheated rollers in fragmented frames from other thermal sources not related to the idlers. In doing so, they claim to have solved the problem of identifying overheated reels in IR images with complex backgrounds. The CM process proposed in the study for the classification of overheated idlers is innovative compared to Classical IRT methods, and the results obtained are sufficient. It is appropriate to publish the relevant work in Sensors. Consequently, it is appropriate to publish the relevant work in Sensors. The following minor edits should be considered prior to publication.

  1. Some typos should be corrected. For example, "sued" on page 13 line 334.

Thanks for your kind reminders. We have addressed the mentioned issue

  1. All symbols in the manuscript should be reviewed. Symbols used in the text should be shown in the same way as those used in equations. For example, "P(g)" on page 8 line 235, "L" on line 238, "T" on line 265.

Thank you very much for the reminder. We have reviewed the manuscript and corrected the symbols.

  1. Figure representations should be the same throughout the manuscript. For example, "Fig11" on page 14 line 366.

Thank you for the nice reminder. We have addressed the mentioned issue in the manuscript.

  1. Reference notations in the text should be arranged. For example, the reference notation in the sentence on page 11 line 306 is "The parameters such as environmental conditions and emissivity values of the objects should be taken into account for accurate identification of overheated equipment.[32]."

Thanks for your comment. We have proofread the manuscript and corrected the mirror errors.

  1. Some minor visual errors in the figures should be corrected. For example, Figure 9 should be rearranged (output text in frame).

Thanks for your kind reminders. We checked the mentioned Figure and edited the shapes as requested.

  1. The phrase "10" on page 14 line 344 should be revised.

Thank you very much for pointing this out. We address the mentioned error on the manuscript.

Round 2

Reviewer 1 Report

The authors have responded to the comments properly. The manuscript can be accepted now.